# Fermentation Quality, In Vitro Digestibility, and Aerobic Stability of Ensiling Spent Mushroom Substrate with Microbial Additives

**DOI:** 10.3390/ani13050920

**Published:** 2023-03-03

**Authors:** Qixuan Yi, Peng Wang, Hongyu Tang, Meng Yu, Tianyue Zhao, Ziyang Sheng, Hailing Luo

**Affiliations:** 1Department of Animal Science, Jilin University, Changchun 130062, China; 2Sanya Institute of China Agricultural University, Sanya 572025, China; 3State Key Laboratory of Animal Nutrition, College of Animal Science and Technology, China Agricultural University, Beijing 100193, China

**Keywords:** additives, aerobic stability, fermentation quality, in vitro digestibility, spent mushroom substrate

## Abstract

**Simple Summary:**

In animal husbandry, spent mushroom substrate has not been effectively utilized as an agricultural byproduct. Ensiling is an important roughage processing method that can not only improve the palatability of feed but also prolong the storage time of high-water feed. At present, due to the different physicochemical properties of various agricultural byproducts, a unified conclusion on the effects of using microbial additives in agricultural byproducts has not been reached. Therefore, this experiment investigated the effects of lactic acid bacteria and cellulase on the fermentation quality, in vitro digestibility, and aerobic stability of *Flammulina velutipes* spent mushroom substrate silage (F-silage) and *Pleurotus eryngii* spent mushroom substrate silage (P-silage). It was found that the combination of application of lactic acid bacteria and cellulase is an effective strategy for improving the fermentation quality and aerobic stability of F-silage and P-silage. In addition, cellulase can be used as an additive to improve the in vitro digestibility of P-silage. Our research results provide technical support for a thorough understanding of the effects of microbial additives on spent mushroom substrate silage and provide a theoretical basis for the production of high-quality spent mushroom substrate fermented feed.

**Abstract:**

This experiment investigated the effects of lactic acid bacteria and cellulase on the fermentation quality, in vitro digestibility, and aerobic stability of *Flammulina velutipes* spent mushroom substrate silage (F-silage) and *Pleurotus eryngii* spent mushroom substrate silage (P-silage). Silage treatments included groups without any additives (control), with lactic acid bacteria (L), with cellulase (E), and with lactic acid bacteria and cellulase (M). Data analysis was performed using independent sample *t*-test and analysis of variance. After 45 days of ensiling, the pH in F-silage and P-silage from the L, E, and M groups were lower than those in the control group (*p* < 0.05). The pH, acetic acid (AA), and propionic acid (PA) levels in P-silage were lower than those in F-silage, and the LA content in P-silage was higher than that in F-silage (*p* < 0.05). Compared with the control, the E treatment increased in vitro neutral detergent fibre digestibility (IVNDFD) and in vitro acid detergent fibre digestibility (IVADFD) in F-silage and P-silage (*p* < 0.05). The aerobic stability of F-silage inoculated with L increased (*p* < 0.05) by 24 h compared to the control. The aerobic stability of P-silage inoculated with M increased (*p* < 0.05) by 6 h compared to the control. The improvement in fermentation quality and aerobic stability is extremely large in terms of applying M in F-silage and P-silage. The E is effective in improving the in vitro digestibility of P-silage. The research results provide a theoretical basis for the production of high-quality spent mushroom substrate fermented feed.

## 1. Introduction

With the increase in feed prices, the beef cattle-breeding industry is facing great production cost pressures. Rising feed costs also herald a large emphasis on underutilized and inexpensive agricultural byproducts. Agricultural byproducts can be composted to form natural organic fertilizers or used as soil conditioners to improve soil structure, but the utilization rate of agricultural byproducts is not high. The optimal process is to prepare agricultural byproducts into animal feed, thereby enhancing the sustainability of feed resources and alleviating the shortage of roughage resources in production [1].

Spent mushroom substrate (SMS), which usually uses agricultural byproducts, including corncob, wheat bran, rice straw, and residues, such as raw materials for edible fungi cultivation, is the base residue of edible fungi culture medium after harvesting [2]. Edible fungi not only absorb nutrients in the culture medium to maintain their growth but also degrade certain fibrous substances in the culture medium through cellulose complex enzymes and peroxidase. There are many edible fungal mycelial residues in SMS that are complexes composed of amino acids, organic acids, and other nutrients produced by the enzymatic conversion of cellulose and biologically active substances [3]. The agricultural byproducts of high fibre crops, such as straw, corncob, cottonseed shell, sawdust, and bran, can provide more abundant feed sources and large amounts of energy for ruminants. For example, most SMS raw materials are agricultural byproducts of high fibre crops, which have low nutritional value, poor palatability, low digestibility and absorption, and low feeding value. However, the nutritional value and palatability of SMS raw materials can be improved through microbial fermentation, thus improving the production performance of ruminants.

Approximately 39.34 million tons of edible fungi were produced in China in 2019, accounting for more than 85% of the global total (China Edible Fungi Association 2020). Under normal conditions, 1 kg of edible fungi will produce 5 kg of SMS [4]. According to this ratio, 197 million tons of fresh SMS could be produced in China in 2019. In China, the production of edible fungi is mainly divided into two production modes: family small-scale and factory standardization. Due to factors such as scattered production areas, complex culture medium composition, and unstable yield and quality, most of the SMS produced by family small-scale are treated by in situ disposal or nonfuel incineration, which has caused serious environmental pollution problems. Although the SMS produced by the factory standardization has stable nutrients and high feed value, it is mainly used in power plants to generate electricity, and only a small amount is used for beef cattle fattening. Therefore, to protect the environment and promote the sustainable development of recycling-based animal husbandry, it is necessary to develop effective SMS feed preparation and storage technology.

The moisture content in SMS is generally between 50% and 75%, which is not easy to store. Ensiling is an important roughage processing method that can not only improve the palatability of feed but also prolong the storage time of high-water feed and maintain the original nutrient level [5]. Biotransformation in the process of ensiling can delay the spoilage of SMS and can convert it into animal feed resources with high economic benefits. Lactic acid bacteria preparation is effective in improving the fermentation quality of cassava residue silage [6]. During ensiling, the metabolites produced by lactic acid bacteria can inhibit the growth of Clostridium, moulds, and other harmful bacteria. The water-soluble carbohydrate content in SMS is low. Therefore, cellulase can be used to promote the growth of lactic acid bacteria. Cellulase promotes the decomposition of the plant cell wall, degrades plant fibres into monosaccharides, and provides a sufficient fermentation substrate for lactic acid bacteria growth [7]. 

At present, due to the different physicochemical properties of various agricultural byproducts, there is no unified conclusion on the effects of using microbial additives in agricultural byproducts [8]. According to the previous research results of soybean residue corn straw mixed silage [9], we speculate that the combined application of lactic acid bacteria and cellulase can increase the lactic acid content of SMS silage, delay the deterioration of SMS silage, and further improve the fermentation quality, in vitro digestibility, and aerobic stability of SMS silage. Therefore, to provide a theoretical basis for the production of high-quality SMS fermented feed, the effects of lactic acid bacteria and cellulase on the fermentation quality, in vitro digestibility, and aerobic stability in the industrially produced SMS silage were investigated in this experiment.

## 2. Materials and Methods

### 2.1. Silage Production and Treatments

The experiment was conducted at the Animal Nutrition and Feed Science Experimental Base of Jilin University, Changchun City, Jilin Province, China (43.88° N, 125.35° E, above sea level 300 m). The *Flammulina velutipes* spent mushroom substrate (F-SMS) and *Pleurotus eryngii* spent mushroom substrate (P-SMS) used in this experiment were the base residue of edible fungi culture medium after harvesting. The raw material composition of the two edible fungal cultivation media is shown in Table 1.

The experimental treatments included (1) no additive (control); (2) lactic acid bacteria (L, strain number of the *L. plantarum*: *Lactobacillus plantarum* LP1, provided by SNOW BRAND SEED Co., Ltd., Chikusou-1) applied at 50 mg kg^−1^ FW, and the actual number of lactic acid bacteria was 4.7 × 10^7^ colony forming units (CFUs) g^−1^ FW; (3) cellulase (E, utilizing *Trichoderma viride* and *Acremonium cellulolyticus* Y-94) applied at 50 mg kg^−1^ FW, and the actual enzymatic activity was 4.2 × 10^−3^ U g^−1^ FW; and (4) a mixed preparation of lactic acid bacteria and cellulase (M). The amount of lactic acid bacteria and cellulase added was 50 mg kg^−1^ FW. The actual number of lactic acid bacteria was 4.7 × 10^7^ CFUs g^−1^ FW, and the actual enzymatic activity was 4.2 × 10^−3^ U g^−1^ FW.

After the fresh SMS was treated with additives, it was placed into a 5 L polyethene plastic bucket with an inner lid (five 5-L-polyethylene-plastic-bucket drums per treatment). The packing densities (mean ± standard deviation) of F-SMS and P-SMS were 550.1 ± 10.0 kg/m^3^ FW and 575.0 ± 9.2 kg/m^3^ FW, respectively. The fermentation barrels were sealed, weighed, and stored at ambient temperatures (21–25 °C) for 45 days to prepare *F. velutipes* spent mushroom substrate silage (F-silage) and *P. eryngii* spent mushroom substrate silage (P-silage). Five replicates per treatment were used for the in vitro digestibility analysis. Three replicates were used for the fermentation quality analysis and the other replicates were mixed and used for the aerobic stability test.

### 2.2. Chemical Composition Analysis

The silage samples were dried in a forced-air oven at 60 °C for 72 h and then ground and sieved through a 1 mm sieve for chemical analysis. Dry matter (DM), organic matter (OM), ether extract (EE), ash, crude fibre (CF), and crude protein (CP) contents were determined using the method from the Association of Official Analytical Chemists [10]. Neutral detergent fibre (NDF), acid detergent fibre (ADF), and acid detergent lignin (ADL) contents were determined using the procedure reported by Van Soest et al.—the NDF and ADF were determined sequentially using neutral detergent and acid detergent, respectively [11]. Buffering capacity (BC) was measured by the method of Playne and McDonald [12]. Calcium (Ca) and total phosphorus (P) were determined by the potassium permanganate method and the ammonium molybdate spectrophotometric method, respectively [10]. Water-soluble carbohydrate (WSC) was determined by the sulfuric acid-anthrone colorimetric method [13]. 

Gross energy (GE) was determined using an oxygen bomb calorimeter [10]. The digestible energy (DE), metabolizable energy (ME), net energy for maintenance (NEm), net energy for weight gain (NEg), and net energy for lactating cows (NEl) were calculated according to the method of Gao et al. [14]. The following formula were used for the calculations.
DE = GE × [70.19 − 1.364 × (ADF − 29.83) + 0.104 × CP + 0.149 × EE + 0.022 × NDF − 0.244 × ash]/100(1)
ME = DE × [86.38 − (9.9 × CF + 19.6 × CP)/(100 − ash)]/100(2)
NEm = ME × (0.287 × ME/GE + 0.554)(3)
NEg = ME × (0.78 × ME/GE + 0.006)(4)
NEl = ME × [0.6 + 0.24 × (ME/GE − 0.57)](5)
Notes: (1) The units of GE, DE, ME, NEm, NEg, and NEl are MJ/kg DM. (2) The unit of CF, CP, EE, NDF, ADF, and ash is g/kg of DM.

Formula (1) was derived from 347 experimental data in different countries. The CP content in the feed was 90–350 g/kg, with an average content of 175 g/kg. The CF content in the feed was 150–430 g/kg, with an average content of 256 g/kg. In addition, Formulas (2)–(5) were derived based on Formula (1).

### 2.3. Fermentation Quality and Microbiological Analyses

A total of 20 g of sample was diluted with 180 mL of distilled water; then, the solution was stirred until homogenous and stored in the refrigerator at 4 °C for 24 h. Thereafter, the mixture was filtered through quantitative filter paper [15]. The pH was determined by a pH meter (PHSJ-4F, Yidian Scientific Instrument Co., Ltd., Shanghai, China). Ammonia nitrogen (AN) was determined by steam distillation [16]. Lactic acid (LA), acetic acid (AA), propionic acid (PA), and butyric acid (BA) were determined by high-performance liquid chromatography (Column: Shodex RS Pak KC-811, Showa Denko KK, Kawasaki, Japan; Detector: DAD, 210 nm, SPD-20A, Shimadzu Co., Ltd., Kyoto, Japan; mobile phase: 3 mmol/L HClO_4_; flow rate: 1 mL/min; detection wavelength: 210 nm; temperature: 50 °C; injection volume: 10 µL). 

The fresh sample (20 g) was evenly mixed with 180 mL of sterile physiological saline (0.85% NaCl), shaken for 30 min on a shaker, and then sequentially diluted in sterile physiological saline according to a 10^−1^–10^−6^ gradient. The three best dilutions (10^−3^–10^−5^) for determining the bacterial colony count were selected. Finally, 100 μl dilutions of different concentrations were spread evenly on the agar medium described below. Lactic acid bacteria were counted on De Man, Rogosa, and Sharp agar medium (Budweiser Technology Co., Ltd., Shanghai, China) and incubated at 37 °C for 48 h under anaerobic conditions. Aerobic bacteria were counted on nutrient agar medium (Hope Biotechnology Co., Ltd., Qingdao, China) and incubated at 37 °C for 48 h. Yeast and moulds were counted on potato dextrose agar medium (Budweiser Technology Co., Ltd., Shanghai, China) and incubated at 28 °C for 48 h. The number of microorganisms was counted on plates yielding 20–200 CFUs. All microbiological data were converted to log_10_ CFUs g^−1^, and the results are reported as fresh weight [17].

### 2.4. Aerobic Stability Analysis

Aerobic stability was determined based on the time required for the silage temperature to be 2 °C higher than the ambient temperature [18]. The 1 kg sample participating in the aerobic stability test was put into a clean plastic bucket with a volume of 1 L, and the fermented silage was exposed to aerobic exposure at 23.5 °C ± 1 °C for 14 days. A thermometer was placed in the centre of the silage, and the temperature was recorded every 6 h.

### 2.5. In Vitro Degradability Analysis

This study strictly followed the Chinese Laboratory Animal Welfare Ethical Review Guidelines and was approved by the Laboratory Animal Welfare Ethics Committee of Jilin University (Number of permit: SY202009600). A 0.5 g sample was taken and put into a filter bag (ANKOM F57; aperture 25 um; ANKOM Technology Corporation; Macedon, NY, USA), which was heat-sealed and placed in a 130 mL serum bottle. In the presence of CO_2_, rumen fluid (collected from four sheep fed alfalfa and corn silage) was filtered through four layers of cheesecloth and mixed in equal volumes. The mixed filtrate was mixed with McDougall’s artificial saliva at a ratio of 1:4 (*v*/*v*). A 60 mL mixture was placed into a 130 mL serum bottle, rinsed with CO_2_, and incubated at 39 °C for 72 h in a water bath. After 72 h of incubation, the filter bag was removed from the serum bottles and rinsed with cold distilled water. The filter bag was dried in a forced air drying oven at 100 °C for 24 h, and the residue was weighed to determine the in vitro dry matter digestibility (IVDMD) [7]. In vitro neutral detergent fibre digestibility (IVNDFD) and in vitro acid detergent fibre digestibility (IVADFD) were determined, respectively, by analysing the content of residual NDF and ADF in the residue [19]. Hemicellulose (HC) = NDF-ADF (the NDF and ADF run sequentially on the same original sample). IVDMD, IVNDFD, IVADFD, and in vitro hemicellulose digestibility (IVHCD) are calculated as follows: (respective weights of DM, NDF, ADF and HC before digestion—respective weights of DM, NDF, ADF and HC remaining)/(respective weights of DM, NDF, ADF and HC before digestion).

### 2.6. Statistical Analyses

All data were analysed by SPSS statistical software (version 26; International Business Machines Corporation; Armonk, NY, USA). An independent sample *t*-test was used to analyse the significant differences between F-silage and P-silage for each item. One-way analysis of variance (ANOVA) was used to examine the significant differences between the mean values of the fermentation quality, chemical composition, energy, aerobic stability, and in vitro digestibility of the silage. Based on the results from the significance test, Tukey’s method was used for multiple comparisons. Lactic acid bacteria (L) and cellulase (E) were used as fixed factors, and a two-way analysis of variance was used to evaluate the main effects of each factor and interactions on fermentation quality, chemical composition, energy, and in vitro digestibility. 

## 3. Results

The chemical composition, BC, microbial population, and energy of F-SMS and P-SMS before ensiling are shown in Table 2. The DM content of P-SMS was 12 g kg^−1^ FW greater than that of F-SMS and the WSC content was 1.4 times higher than that of F-SMS. However, the BC value of F-SMS was 1.8 times higher than that of P-SMS. The epiphytic numbers of lactic acid bacteria in F-SMS and P-SMS were 4.68 log_10_ CFUs g^−1^ FW and 4.30 log_10_ CFUs g^−1^ FW, respectively. Except for NDF and Ca, there were significant differences between F-SMS and P-SMS in terms of the other indicators (*p* = 0.000–0.001).

### 3.1. Fermentation Quality of Spent Mushroom Substrate Silage

The fermentation quality of F-silage and P-silage is shown in Table 3. The pH, AA, and PA in P-silage were lower than those in F-silage (*p* = 0.000–0.041), and the LA content was higher than that in F-silage (*p* = 0.000–0.020). The L and E treatments influenced pH in F-silage (*p* = 0.002–0.005). The L treatment and the interaction (L × E) influenced LA content in F-silage (*p* = 0.002) and pH in P-silage (*p* = 0.000–0.001). The E treatment influenced pH in P-silage (*p* = 0.000). The L treatment influenced ammonia nitrogen/total nitrogen (AN/TN) ratio in P-silage (*p* = 0.008).

The pH in F-silage and P-silage from the L, E, and M groups was lower than those in the control group (*p* < 0.05). The LA content in F-silage from the L, E, and M groups was higher than that in the control group (*p* < 0.05). The AN/TN in F-silage from the M group was lower than that in the control group (*p* < 0.05). The AN/TN in P-silage from the L group was lower than that in the control group (*p* < 0.05).

### 3.2. Chemical Compositions of Spent Mushroom Substrate Silage

The chemical compositions of F-silage and P-silage are shown in Table 4. The DM, ADF, and ADL contents in F-silage were lower than those in P-silage (*p* = 0.000–0.040), and the CP content was higher than that in P-silage (*p* = 0.001–0.009). The E treatment influenced ADL content in P-silage (*p* = 0.001). The interaction (L × E) influenced DM and NDF contents in P-silage (*p* = 0.002–0.007).

The DM content in P-silage from the L, E, and M groups were higher than that in the control group (*p* < 0.05). The ADF content of P-silage treated with L was lower than that in the control (*p* < 0.05). The CP and NDF contents in P-silage from the M group was higher than those in the control group (*p* < 0.05). Compared with the control, the E and M treatments produced lower ADL contents in P-silage (*p* < 0.05).

### 3.3. Energy and In Vitro Digestibility of Spent Mushroom Substrate Silage

The energy and in vitro digestibility of F-silage and P-silage are shown in Table 5 and Table 6. The DE, ME, NEm, NEI, and NEg in F-silage were higher than those in P-silage (*p* = 0.000–0.011), and the IVDMD, IVADFD, and IVNDFD were higher than those in P-silage (*p* = 0.000–0.017). The L treatment and the interaction (L × E) influenced IVHCD in F-silage (*p* = 0.002–0.007). The E treatment and the interaction (L × E) influenced IVNDFD in F-silage (*p* = 0.000–0.020). The E treatment influenced IVADFD in F-silage (*p* = 0.000). The L treatment influenced IVHCD in P-silage (*p* = 0.012).

Compared with the control, the E and M treatments produced higher IVNDFD and IVADFD in F-silage (*p* < 0.05). The L treatment resulted in higher IVDMD of P-silage (*p* < 0.05), while the L treatment produced a lower IVHCD of F-silage (*p* < 0.05). The E treatment increased IVNDFD and IVADFD in F-silage and P-silage (*p* < 0.05). Compared with the control, the L and E treatments produced higher IVHCD in P-silage (*p* < 0.05).

### 3.4. Aerobic Stability of Spent Mushroom Substrate Silage

The changes in temperature of F-silage and P-silage aerobic exposure are shown in Figure 1 and Figure 2. The temperature of all treatments in F-silage increased within 24–96 h, but the temperature increase range was small (Figure 1). Compared with F-silage, the temperature fluctuation in P-silage was more obvious (Figure 2). After 72 h, the temperature of the L and E treatments began to increase evidently. After 144 h, the temperature in the L treatment began to decrease. After 120 h, the temperature in the control and M treatment groups began to obviously increase.

The aerobic stability of F-silage inoculated with L and M increased (*p* < 0.05) by 24 h compared to the control; additionally, inoculation with E increased (*p* < 0.05) the aerobic stability of F-silage by 18 h (Figure 3). The aerobic stability of P-silage inoculated with M increased (*p* < 0.05) by 6 h compared to the control (Figure 4). The aerobic stability of P-silage inoculated with L reduced (*p* < 0.05) by 42 h compared to the control; additionally, inoculation with E reduced (*p* < 0.05) the aerobic stability of P-silage by 48 h (Figure 4).

## 4. Discussion

In the introduction, we speculated that the combined application of lactic acid bacteria and cellulase is able to increase the lactic acid content of SMS silage, delay the deterioration of SMS silage, and further improve the fermentation quality, in vitro digestibility, and aerobic stability of SMS silage. From Table 3, F-silage and P-silage obtained good fermentation, and the fermentation quality of M treatment group was the best, which was consistent with our previous speculation. From Table 6, for F-silage, the IVDMD, IVNDFD, and IVADFD of the M treatment group was the highest. However, the M treatment group did not affect the in vitro digestibility of P-silage. From Figure 3 and Figure 4, in F-silage and P-silage, the aerobic stability of the M treatment group was the best, which was consistent with our previous speculation.

### 4.1. Effects of Additives on Fermentation Quality

In general, high-quality silage has a pH of 3.8–4.2, at which point the activity of harmful bacteria in the silage is attenuated [20]. In this study, the pH of the L, E and M treatment groups of P-silage was 3.93–3.96, the LA content was higher, and the AN/TN ratio was lower, indicating that P-silage obtained good fermentation. The fermentation effect of F-silage was lower than that of P-silage, as reflected in the high pH (4.2–4.4) and BA (5.0 g kg^−1^ DM−11.2 g kg^−1^ DM), despite the higher LA content than well-preserved silages (30 g kg ^−1^ DM) [21]. In the presence of a sufficient LA content, the high water content and BC value (282 mEq kg^−1^ DM) of the F-SMS feedstock were also important factors inducing low-quality fermentation [8,14]. High pH may also affect the activity of the proteolytic *Clostridium* and *Enterobacter* bacteria, causing secondary fermentation [22].

The number of lactic acid bacteria on the feed surface must be at least 5 log_10_ CFUs g^−1^ FW for the silage to be fermented with good quality. The number of lactic acid bacteria in the two SMS was measured to be < 5 log_10_ CFUs g^−1^ FW before silage, and thus a lactic acid bacteria preparation was needed to improve silage fermentation. Studies have shown that adding *L. plantarum* during ensiling effectively improves the fermentation quality of *P. ostreatus* spent mushroom substrate silage [23]. The *L. plantarum* reduced the pH and BA content of *Leymus chinensis* silage, while increasing the LA content [24]. Compared with the control group, the addition of *L. plantarum* accelerated the accumulation of LA, decreased the pH and AN/TN ratio, and improved the fermentation quality of forage or alfalfa silage, which was consistent with the results of this experiment in F-silage [25,26]. Structural polysaccharides in plant cell walls are decomposed into monosaccharides by cellulase, which provide sufficient substrates (fermentable sugars) for lactic acid bacteria fermentation, thereby improving fermentation quality [27]. Zhao et al. [9] showed that adding cellulase reduced the pH and AN/TN of bean dregs and corn stover mixed silage, and improved the fermentation quality of silage, consistent with the experimental results obtained for P-silage. Compared to the control group, the mixed addition of *L.* plantarum and cellulase significantly reduced the pH of corn stover silage, and the pH of the mixed addition treatment group was minimal, which was consistent with the results of this experiment [14]. The AN produced by *Clostridium* was also an important indicator for evaluating the quality of silage fermentation, reflecting the degradation level of proteins and amino acids during ensiling [28]. The AN/TN ratio of F-silage and P-silage in each additive treatment group was lower than the control treatment. The explanation was that a substantial decrease in pH inhibits plant enzymes and aerobic microorganisms, reducing protein degradation [28]. The concentration of AA in both the L treatment and M treatment groups of P-silage was lower than that in the control treatment group. This result may be due to the higher fermentation efficiency of silage due to the exogenous addition of lactic acid bacteria, while the fermentation efficiency of the lactic acid bacteria attached to the surface of the feed material was lower and the pH decreased more slowly [29]. This hypothesis was supported by the experiment reported by Silva et al. [22], who found that the AA concentration of alfalfa silage inoculated with lactic acid bacteria was lower than that of the control treatment group because the lactic acid bacteria attached to the feed surface in the control treatment group were less efficient in inducing rapid acidification in the early fermentation stage. This finding was in stark contrast to the results reported by Filya [30] that the AA concentration of low dry matter maize silage inoculated with heterofermentative lactic acid bacteria (*Lactobacillus brucei*) was significantly higher than that of control and *L. plantarum* treatments. This difference may be attributed to the difference in AA concentration caused by the different fermentation types mediated by lactic acid bacteria. The metabolism of homofermentative lactic acid bacteria produces only LA, while the metabolism of heterofermentative lactic acid bacteria produces higher contents of AA and various metabolites. The significant difference in the AA content of F-silage and P-silage in this experiment might also be attributed to the different types of lactic acid bacteria populations that epigenetically grow on the surface of feed ingredients. The M treatment of P-silage increased the LA content by 3.3–6.8% and decreased the pH by 0.02–0.28 units compared with the control and L and E treatments. The reason for this was that *L. plantarum* and cellulase were added at the same time, and the two exerted a synergistic effect. Cellulase promotes the degradation of fibrous substances, provides an additional fermentation substrate for lactic acid bacteria, and helps lactic acid bacteria dominate the microbial community in silage such that the fermentation moves towards homogenous fermentation [31]. Therefore, in this study, the addition of M was suggested to improve the quality of P-silage.

### 4.2. Effects of Additives on Chemical Composition and In Vitro Digestibility

Compared with the two types of SMS raw materials, the content of OM did not change significantly in the silage of the two types of SMS in this experiment, and the CP content was increased, indicating that the two types of SMS better preserve their nutritional components. This increase may be due to the rapid inhibition of proteolysis or a concentration effect caused by organic carbon loss during ensiling [32]. High-quality silage fermentation reduces protein hydrolysis in non-protein nitrogen, ammonia nitrogen, and other non-protein compounds and generally retains the nutrients of raw silage materials [33]. The CP content is also an important indicator for evaluating the nutritional value of silage. The higher the CP content, the better the feed value. Compared with the control treatment, better protein preservation was observed in alfalfa and corn silage inoculated with homofermentative lactic acid bacteria, indicating that the addition of lactic acid bacteria may reduce the loss of nutrients to a certain extent, consistent with the test results for the L treatment of P-silage [34]. Compared with the control treatment, E or M treatment decreased the NDF, ADF, and ADL contents in F-silage; increased the CP content; and improved the feed value of F-silage. The explanation for this result was that cellulase decomposed the fibre and other components of F-silage plant cell wall, fully releasing the cell contents, increasing the available WSC content for lactic acid bacteria, and reducing the consumption of CP [9]. The AN concentration is one of the key indicators to determine the degree of protein hydrolysis in silage. The extensive hydrolysis of proteins will affect nitrogen utilization by ruminants. The smaller AN/TN ratio also proved the good effect of E and M treatment on preserving proteins from another aspect. Dehghani et al. [35] showed that adding cellulase directly hydrolysed the fibrous material in silage, which was the most effective at reducing the NDF content of corn stover silage, consistent with the results of our experiment. The contents of NDF, ADF, and ADL in the L treatment group of P-silage were lower than in the control group, and this result was surprising because lactic acid bacteria do not directly decompose plant fibres and lignin [36]. Some microorganisms might degrade fibre and lignin during the silage process, which may explain the decrease in fibre and lignin contents in silage. The chemical composition of the silage determines the nutritional value of the livestock. Dry matter loss in silage is associated with the depletion of soluble carbohydrate and organic acid components [37]. In P-silage, the DM content in L, E, and M treatment groups was significantly higher than that in the control group, indicating that the DM loss of P-silage was reduced after the additive was applied, and the P-silage nutrients were better preserved because of the higher WSC content in the P-SMS raw material. Therefore, during the fermentation process of lactic acid bacteria, WSC, a fermentation substrate, was metabolized into organic acids, thereby reducing the loss of nutrients [38]. The study showed that about 2% of the DM content were consumed through the action of the additive [39]. After 45 days of ensiling, the DM content of both silages was lower than the DM content of the raw material.

In general, the energy distribution potential of the feed was mainly assessed by determining the IVDMD in ruminant fluids of ruminants [40]. The IVDMD and IVNDFD of F-silage were higher than those of P-silage. This difference was related to the difference in carbohydrate types between the two SMS feedstocks (Table 1). The IVDMD of F-silage in the E and M treatment groups were slightly higher than that in the control treatment group. A potential explanation for this result was the enzymatic saccharification of exogenously added cellulase, where sugars and proteins were enzymatically generated to glycoproteins, increasing the rumen-fermentable fraction of silage [20]. The experimental results reported by Rinne et al. [41] support this result. The enzymatic saccharification reaction of cellulase contributes to lactic acid fermentation, thereby increasing the in vitro OM digestibility of the silage. The additive treatment also increased the digestible and metabolizable energy of F-silage. Bala at al. [42] added an enzyme preparation containing cellulase to the diet of goats and found that IVDMD, IVNDFD, and IVADFD were increased, indicating that cellulase effectively improved the digestibility of cellulose feed in the rumen. The reason for this might be that cellulase could destroy glycosidic bonds and disintegrate the cell wall structure of silage, thus reducing the resistance of the cell wall to rumen microorganisms and improving the in vitro digestibility of silage, which was consistent with the results of F-silage. The L treatment increased the IVDMD of P-silage compared to the control treatment, due to the effect of lactic acid bacteria on reducing dry matter loss during silage fermentation. Lactic acid bacteria improved the fermentation quality of P-silage, inhibited the digestion and hydrolysis of protein, and subsequently increased IVDMD. The addition of lactic acid bacteria to vegetable residue silage, such as cabbage and lettuce, increases the IVDMD of the silage, which is consistent with the aforementioned experimental results [43]. The fermentation effect of F-silage was lower than that of P-silage, as reflected in the high pH (4.2–4.4) and BA (5.0 g kg^−1^ DM−11.2 g kg^−1^ DM). If F-silage treated with additive is introduced into a sheep’s diet, it affects the food intake of the sheep and will very likely to lead to weight loss due to poor feed and fermentation quality. The barley silage ration inoculated with cellulase (0.75 mL/kg) improved milk production efficiency, milk fat-corrected milk yield, and the feed digestibility of dairy cows [44]. Therefore, if the E-treated P-silage is introduced into a sheep’s diet, it may maximize the use of agricultural byproduct, promote the balance of the animal’s diet, and improve the production performance of the sheep.

### 4.3. Effect of Additives on Aerobic Stability

The aerobic spoilage of fermented feed is mainly caused by the proliferation of harmful microorganisms, such as yeast and mould [45]. After the silage is opened, the anaerobic environment is destroyed, and the yeast becomes active, which breaks down lactic acid and protein, releases carbon dioxide and ammonia nitrogen, and causes a loss of nutrients. Due to the depletion of LA, the pH of the silage will increase, causing the rapid proliferation of moulds that spoil the silage. AA and PA, acting as inhibitors of aerobic microorganisms, improve the aerobic stability of silage, which increases exponentially with the concentration of AA [46]. In F-silage, L, E, and M treatments all contained high AA concentrations and the aerobic stability increased at 24 h, 18 h, and 24 h, respectively (Figure 3). Compared with F-silage, P-silage contained very low levels of AA and PA, which was more likely to cause aerobic spoilage.

If the silage maintains a high aerobic stability, it will greatly preserve the nutrition of the silage and reduce the accumulation of mycotoxins [17]. The LA and WSC are substrates for the growth and reproduction of aerobic microorganisms, such as yeast and mould [47]. Silage additives such as *L. plantarum* or cellulases also reduced the aerobic stability of silage [48]. For example, the addition of *L. plantarum* did not inhibit the reproductive growth of yeast, resulting in the instability of maize and alfalfa silage during aerobic exposure [49,50], consistent with the results from the P-silage test. The potential explanation was that the L treatment had a higher LA content and produced fewer short-chain fatty acids that inhibited the growth and reproduction of yeasts and moulds on P-silage. In addition, the addition of *L. brucei* reduced the number of harmful bacteria in sunflower and sorghum silage and prolonged the spoilage time of the silage, which was in sharp contrast to the results of this experiment [30,51]. At the same time, the addition of E decreased the aerobic stability of P-silage and caused the silage to spoil earlier. The reason was that the rapid accumulation of a fermentation substrate promoted the reproduction and growth of yeasts and moulds, resulting in the secondary fermentation of P-silage [52]. The P-SMS raw material contained a higher WSC content, which provided a fermentation substrate for the growth of undesired bacteria and may be responsible for the reduced aerobic stability of P-silage treated with L and E (Figure 4). The AA and PA possessed strong antifungal activities and inhibited the growth and reproduction of yeast and mould [21,24]. The aerobic stability of the M treatment group was the best among F-silage and P-silage at 96 h and 138 h, respectively. Therefore, the M treatment was more stable during aerobic exposure and improved the aerobic stability of F-silage and P-silage.

## 5. Conclusions

Overall, the combination of application of lactic acid bacteria and cellulase is an effective strategy for improving the fermentation quality and aerobic stability of F-silage and P-silage. Cellulase can be used as an additive to improve the in vitro digestibility of P-silage. However, in vivo studies are needed to validate our findings and examine the silage quality and effect of cellulase-treated P-silage on beef cattle fattening before recommending it to farmers.

## Figures and Tables

**Figure 1 animals-13-00920-f001:**
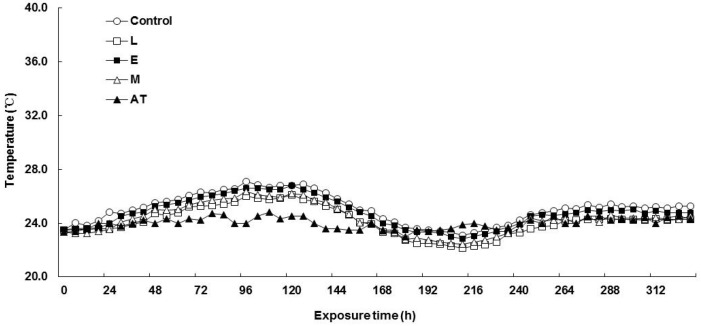
Effect of application of additives on the temperature change of *Flammulina velutipes* spent mushroom substrate silages after exposure to aerobic conditions. L, lactic acid bacteria; E, cellulase; M, the mixture of lactic acid bacteria and cellulase; AT, ambient temperature.

**Figure 2 animals-13-00920-f002:**
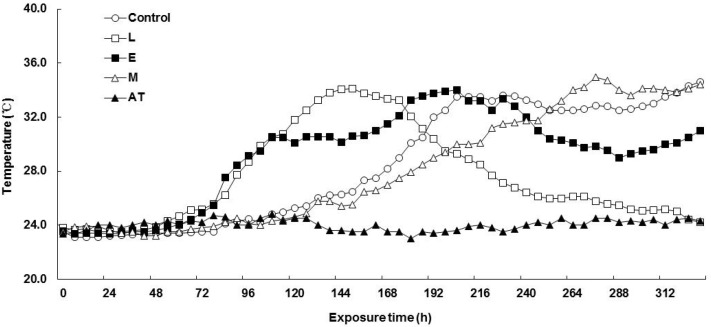
Effect of application of additives on the temperature change of *Pleurotus eryngii* spent mushroom substrate silages after exposure to aerobic conditions. L, lactic acid bacteria; E, cellulase; M, the mixture of lactic acid bacteria and cellulase; AT, ambient temperature.

**Figure 3 animals-13-00920-f003:**
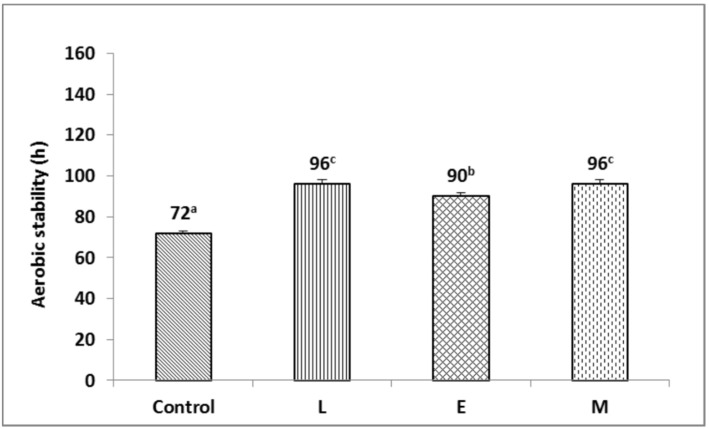
Time required for *Flammulina velutipes* spent mushroom substrate silages temperature to exceed room temperature by 2 °C after exposure to air (one-way ANOVA: F = 117.7, *p* = 0.000). L, lactic acid bacteria; E, cellulase; M, the mixture of lactic acid bacteria and cellulase. ^a–c^ Means with different superscripts differed significantly (*p* < 0.05).

**Figure 4 animals-13-00920-f004:**
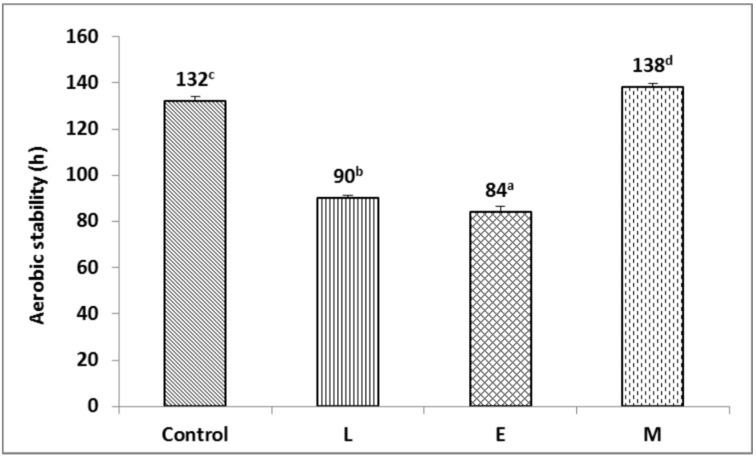
Time required for *Pleurotus eryngii* spent mushroom substrate silages temperature to exceed room temperature by 2 °C after exposure to air (one-way ANOVA: F = 545.5, *p* = 0.000). L, lactic acid bacteria; E, cellulase; M, the mixture of lactic acid bacteria and cellulase. ^a–d^ Means with different superscripts differed significantly (*p* < 0.05).

**Table 1 animals-13-00920-t001:** Percentage of raw material in spent mushroom substrates.

*Flammulina velutipes*		*Pleurotus eryngii*	
Corncob meal (%)	37	Corncob meal (%)	23
Rice bran (%)	38	Corn straw powder (%)	18
Sawdust (%)	8	Sawdust (%)	39
Soybean hulls (%)	5		
Cottonseed hull (%)	5	Soybean meal (%)	6
Brewer’s grain (%)	5	Corn meal (%)	12
Lime (%)	1	Lime (%)	1
Calcium carbonate (%)	1	Calcium carbonate (%)	1
Total (%)	100	Total (%)	100

**Table 2 animals-13-00920-t002:** Characteristics of spent mushroom substrates.

	Spent Mushroom Substrate ^†^			
Item ^‡^	F-SMS	P-SMS	SEM	*p*-Value
Chemical composition and buffering capacity				
Dry matter (g kg^−1^ FW)	492	504	0.351	0.000
Organic matter (g kg^−1^ DM)	872	903	0.677	0.000
Crude protein (g kg^−1^ DM)	90.4	80.7	0.664	0.000
Neutral detergent fibre (g kg^−1^ DM)	729	727	4.112	0.684
Acid detergent fibre (g kg^−1^ DM)	428	497	4.141	0.000
Acid detergent lignin (g kg^−1^ DM)	111	176	3.164	0.000
Water-soluble carbohydrate (g kg^−1^ DM)	24.4	34.2	0.120	0.000
Calcium (g kg^−1^ DM)	17.0	16.5	0.896	0.609
Phosphorus (g kg^−1^ DM)	7.31	1.82	0.247	0.000
Buffering capacity (mEq kg^−1^ DM)	282	157	12.209	0.001
Energy				
GE (MJ kg^−1^ DM)	19.2	18.8	0.034	0.000
DE (MJ kg^−1^ DM)	10.3	7.69	0.090	0.000
ME (MJ kg^−1^ DM)	8.30	6.18	0.071	0.000
NEm (MJ kg^−1^ DM)	5.59	4.00	0.059	0.000
NEl (MJ kg^−1^ DM)	4.69	3.35	0.047	0.000
NEg (MJ kg^−1^ DM)	2.82	1.63	0.051	0.000
Microbial counts				
Lactic acid bacteria (log_10_ cfu g^−1^ FW)	4.68	4.30	0.032	0.000
Yeasts (log_10_ cfu g^−1^ FW)	3.10	4.20	0.040	0.000
Moulds (log_10_ cfu g^−1^ FW)	0.21	2.33	0.072	0.000

SEM, standard error of the mean; ^†^ F-SMS, spent mushroom substrate from *Flammulina velutipes*; P-SMS, spent mushroom substrate from *Pleurotus eryngii*; ^‡^ FW, fresh weight; DM, dry matter; GE, gross energy; DE, digestible energy; ME, metabolizable energy; NEm, net energy for maintenance; NEl, net energy for lactating cow; NEg, net energy for weight gain; cfu, colony-forming units.

**Table 3 animals-13-00920-t003:** Fermentation quality of spent mushroom substrate silages prepared with lactic acid bacteria and cellulase.

Item ^‡^	S-Silage ^§^	Additives ^†^				SEM	*p*-Value	Significance of Main Effects and Interactions		
		Control	L	E	M			L	E	L × E
pH value	F-silage	4.40 ^b^	4.26 ^a^	4.24 ^a^	4.20 ^a^	0.033	0.002	0.005	0.002	0.072
	P-silage	4.21 ^b^	3.96 ^a^	3.95 ^a^	3.93 ^a^	0.031	0.000	0.000	0.000	0.001
	SEM	0.041	0.027	0.032	0.025					
	*p*-value	0.041	0.000	0.001	0.000					
LA (g kg^−1^ DM)	F-silage	56.9 ^a^	77.2 ^b^	67.4 ^b^	67.9 ^b^	0.318	0.002	0.002	0.802	0.002
	P-silage	118	121	122	126	0.889	0.839	0.596	0.488	0.947
	SEM	0.624	0.198	0.861	0.784					
	*p*-value	0.001	0.000	0.020	0.002					
AA (g kg^−1^ DM)	F-silage	144	151	141	151	1.554	0.887	0.462	0.925	0.886
	P-silage	2.72	0.90	0.94	1.04	0.149	0.583	0.443	0.460	0.393
	SEM	1.004	0.273	1.635	1.059					
	*p*-value	0.000	0.000	0.013	0.000					
PA (g kg^−1^ DM)	F-silage	19.5	20.6	18.3	20.4	0.295	0.850	0.456	0.757	0.793
	P-silage	3.44	4.50	5.91	4.70	0.112	0.249	0.903	0.123	0.191
	SEM	0.134	0.101	0.382	0.158					
	*p*-value	0.000	0.000	0.032	0.001					
BA (g kg^−1^ DM)	F-silage	5.01	5.00	11.2	9.76	0.317	0.184	0.757	0.040	0.757
	P-silage	4.33	4.17	5.52	4.18	0.121	0.638	0.398	0.505	0.505
	SEM	0.110	0.062	0.346	0.308					
	*p*-value	0.577	0.250	0.177	0.210					
AN (g kg^−1^ TN)	F-silage	12.8 ^b^	10.9 ^ab^	10.4 ^ab^	9.40 ^a^	0.085	0.023	0.048	0.011	0.490
	P-silage	11.6 ^b^	8.06 ^a^	9.84 ^ab^	9.92 ^ab^	0.068	0.007	0.008	0.894	0.006
	SEM	0.107	0.103	0.028	0.029					
	*p*-value	0.324	0.050	0.130	0.140					

Different lowercase letters (^a^, ^b^) in the same line indicate that there is a significant difference between different additive treatment groups under the same spent mushroom substrate (*p* < 0.05); the *p*-value in the same column represents the *p*-value of the independent sample *t*-test; SEM, standard error of the mean; ^†^ L, lactic acid bacteria; E, cellulase; M, the mixture of lactic acid bacteria and cellulase; ^‡^ DM, dry matter; LA, lactic acid; AA, acetic acid; PA, propionic acid; BA, butyric acid; AN, ammonia nitrogen; TN, total nitrogen; ^§^ S, spent mushroom substrate; F, *Flammulina velutipes* spent mushroom substrate; P, *Pleurotus eryngii* spent mushroom substrate.

**Table 4 animals-13-00920-t004:** Chemical composition of spent mushroom substrate silages prepared with lactic acid bacteria and cellulase.

Item ^‡^	S-Silage ^§^	Additives ^†^				SEM	*p*-Value	Significance of Main Effects and Interactions		
		Control	L	E	M			L	E	L × E
DM (g kg^−1^ FW)	F-silage	414	414	416	412	0.312	0.705	0.398	0.965	0.446
	P-silage	457 ^a^	462 ^b^	462 ^b^	460 ^b^	0.141	0.015	0.084	0.134	0.007
	SEM	0.322	0.122	0.324	0.100					
	*p*-value	0.000	0.000	0.000	0.000					
OM (g kg^−1^ DM)	F-silage	885	887	880	876	1.184	0.803	0.891	0.382	0.740
	P-silage	899	898	897	899	0.130	0.234	0.218	0.469	0.128
	SEM	1.476	0.358	0.595	0.423					
	*p*-value	0.457	0.033	0.054	0.006					
CP (g kg^−1^ DM)	F-silage	115	113	120	119	0.555	0.612	0.737	0.219	0.928
	P-silage	88.2 ^a^	88.8 ^ab^	92.9 ^ab^	95.3 ^b^	0.216	0.032	0.369	0.006	0.564
	SEM	0.381	0.510	0.500	0.230					
	*p*-value	0.002	0.009	0.006	0.001					
NDF (g kg^−1^ DM)	F-silage	690	694	668	662	1.765	0.262	0.910	0.063	0.689
	P-silage	682 ^ab^	650 ^a^	684 ^bc^	717 ^c^	1.035	0.002	0.944	0.002	0.002
	SEM	0.213	0.855	2.295	1.529					
	*p*-value	0.022	0.007	0.527	0.022					
ADF (g kg^−1^ DM)	F-silage	402	406	393	389	0.886	0.260	0.965	0.067	0.566
	P-silage	464 ^b^	445 ^a^	465 ^b^	478 ^b^	0.469	0.001	0.472	0.001	0.001
	SEM	0.281	0.461	1.134	0.656					
	*p*-value	0.000	0.001	0.003	0.000					
ADL (g kg^−1^ DM)	F-silage	135 ^ab^	150 ^b^	95.3 ^a^	101 ^a^	1.372	0.011	0.327	0.002	0.658
	P-silage	188 ^c^	183 ^bc^	159 ^ab^	156 ^a^	0.819	0.009	0.562	0.001	0.828
	SEM	1.760	0.783	0.930	0.727					
	*p*-value	0.040	0.013	0.002	0.002					

Different lowercase letters (^a^, ^b^ and ^c^) in the same line indicate that there is a significant difference between different additive treatment groups under the same spent mushroom substrate (*p* < 0.05); the *p*-value in the same column represents the *p*-value of the independent sample *t*-test; SEM, standard error of the mean; ^†^ L, lactic acid bacteria; E, cellulase; M, the mixture of lactic acid bacteria and cellulase; ^‡^ FW, fresh weight; DM, dry matter; OM, organic matter; CP, crude protein; NDF, neutral detergent fibre; ADF, acid detergent fibre; ADL, acid detergent lignin; ^§^ S, spent mushroom substrate; F, *Flammulina velutipes* spent mushroom substrate; P, *Pleurotus eryngii* spent mushroom substrate.

**Table 5 animals-13-00920-t005:** Energy of spent mushroom substrate silages prepared with lactic acid bacteria and cellulase.

Item ^‡^	S-Silage ^§^	Additives ^†^				SEM	*p*-Value	Significance of Main Effects and Interactions		
		Control	L	E	M			L	E	L × E
GE (MJ kg^−1^ DM)	F-silage	19.3	19.3	19.3	19.1	0.251	0.813	0.468	0.607	0.778
	P-silage	18.9 ^ab^	18.7 ^a^	18.9 ^ab^	19.0 ^b^	0.058	0.034	0.937	0.025	0.032
	SEM	0.343	0.094	0.060	0.048					
	*p*-value	0.288	0.006	0.002	0.040					
DE (MJ kg^−1^ DM)	F-silage	11.1	10.9	11.3	11.2	0.316	0.664	0.698	0.272	0.773
	P-silage	7.89	7.75	7.74	7.79	0.059	0.115	0.254	0.254	0.054
	SEM	0.297	0.063	0.310	0.134					
	*p*-value	0.007	0.000	0.000	0.001					
ME (MJ kg^−1^ DM)	F-silage	8.82	8.69	8.95	8.98	0.262	0.668	0.787	0.279	0.671
	P-silage	6.27	6.19	6.17	6.20	0.046	0.225	0.380	0.216	0.137
	SEM	0.247	0.039	0.262	0.100					
	*p*-value	0.008	0.000	0.008	0.001					
NEm (MJ kg^−1^ DM)	F-silage	6.04	5.94	6.15	6.19	0.205	0.625	0.807	0.249	0.642
	P-silage	4.07	4.01	4.00	4.01	0.037	0.277	0.461	0.193	0.193
	SEM	0.180	0.031	0.215	0.083					
	*p*-value	0.007	0.000	0.009	0.001					
NEl (MJ kg^−1^ DM)	F-silage	5.05	4.96	5.14	5.17	0.171	0.624	0.810	0.250	0.633
	P-silage	3.40	3.36	3.35	3.36	0.029	0.298	0.446	0.236	0.188
	SEM	0.150	0.024	0.180	0.069					
	*p*-value	0.007	0.000	0.009	0.001					
NEg (MJ kg^−1^ DM)	F-silage	3.19	3.11	3.30	3.35	0.165	0.503	0.890	0.175	0.583
	P-silage	1.66	1.63	1.61	1.62	0.026	0.265	0.356	0.129	0.356
	SEM	0.120	0.028	0.190	0.066					
	*p*-value	0.005	0.000	0.011	0.001					

Different lowercase letters (^a^, ^b^) in the same line indicate that there is a significant difference between different additive treatment groups under the same spent mushroom substrate (*p* < 0.05); the *p*-value in the same column represents the *p*-value of the independent sample *t*-test; SEM, standard error of the mean; ^†^ L, lactic acid bacteria; E, cellulase; M, the mixture of lactic acid bacteria and cellulase; ^‡^ DM: dry matter; GE, gross energy; DE, digestible energy; ME, metabolizable energy; NEm, net energy for maintenance; NEl, net energy for lactating cow; NEg, net energy for weight gain; ^§^ S, spent mushroom substrate; F, *Flammulina velutipes* spent mushroom substrate; P, *Pleurotus eryngii* spent mushroom substrate.

**Table 6 animals-13-00920-t006:** In vitro digestibility of spent mushroom substrate silages prepared with lactic acid bacteria and cellulase.

Item ^‡^	S-Silage ^§^	Additives ^†^				SEM	*p*-Value	Significance of Main Effects and Interactions		
		Control	L	E	M			L	E	L × E
IVDMD (g kg^−1^ DM)	F-silage	516 ^ab^	512 ^a^	522 ^ab^	526 ^b^	4.383	0.034	0.968	0.066	0.568
	P-silage	471 ^a^	487 ^b^	468 ^a^	461 ^a^	3.856	0.000	0.478	0.001	0.001
	SEM	0.218	0.358	0.870	0.504					
	*p*-value	0.000	0.001	0.003	0.000					
IVNDFD (g kg^−1^ DM)	F-silage	445 ^a^	439 ^a^	483 ^b^	495 ^c^	0.437	0.000	0.345	0.000	0.020
	P-silage	409 ^a^	416 ^ab^	430 ^b^	405 ^a^	0.500	0.006	0.040	0.225	0.002
	SEM	0.375	0.469	0.657	0.301					
	*p*-value	0.001	0.008	0.001	0.000					
IVADFD(g kg^−1^ DM)	F-silage	353 ^a^	361 ^a^	431 ^b^	452 ^c^	0.479	0.000	0.003	0.000	0.085
	P-silage	339 ^a^	341 ^a^	355 ^b^	329 ^a^	0.465	0.004	0.006	0.560	0.003
	SEM	0.352	0.340	0.556	0.587					
	*p*-value	0.017	0.004	0.000	0.000					
IVHCD(g kg^−1^ DM)	F-silage	573 ^b^	549 ^a^	560 ^a^	557 ^a^	0.418	0.003	0.002	0.405	0.007
	P-silage	560 ^a^	576 ^b^	589 ^c^	556 ^a^	0.382	0.000	0.012	0.138	0.000
	SEM	0.196	0.430	0.538	0.359					
	*p*-value	0.003	0.003	0.006	0.755					

Different lowercase letters (^a^, ^b^, and ^c^) in the same line indicate that there is a significant difference between different additive treatment groups under the same spent mushroom substrate (*p* < 0.05); the *p*-value in the same column represents the *p*-value of the independent sample *t*-test; SEM, standard error of the mean; ^†^ L, lactic acid bacteria; E, cellulase; M, the mixture of lactic acid bacteria and cellulase; ^‡^ IVDMD, in vitro dry matter digestibility; IVNDFD, in vitro neutral detergent fibre digestibility; IVADFD, in vitro acid detergent fibre digestibility; IVHCD, in vitro hemicellulose digestibility; ^§^ S, spent mushroom substrate; F, *Flammulina velutipes* spent mushroom substrate; P, *Pleurotus eryngii* spent mushroom substrate.

## Data Availability

The authors are solely responsible for the completeness of all data and the accuracy of data analysis in this study. Complete data in this study are available from the corresponding authors upon reasonable request.

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
