# Peer review of "Fermentation Quality, In Vitro Digestibility, and Aerobic Stability of Ensiling Spent Mushroom Substrate with Microbial Additives"

_animals, 2023, doi:10.3390/ani13050920_

Round 1
Reviewer 1 Report (New Reviewer)
This study examined the Fermentation Quality, In Vitro Digestibility, and Aerobic Stability of Ensiling Spent Mushroom Substrate with Microbial Additives. It helps with understanding the effects microbial additives on spent mushroom substrate on silage and provides information on the production of high-quality spent mushroom substrate fermented feed.
There are a lot of byproducts in agriculture that goes to waste and this manuscript provides ways to recycle the waste product into meaningful use in ruminant production. Studies of such nature contributes to sustainable agriculture.
I have a few questions and suggestions for the authors.
Line 17-18: Should it be High-water content of feed?
Line 25: why attractive? I suggest removing it
Line 112: the experiment was conducted at the..........
Line 119 Table 1: I would have expected that you use the same raw materials in each spent mushroom substrate because differences could
arise from using different raw materials which could affect the results for effectiveness of the mushroom type.
Table 3: pH value for F-silage was significant p=0.002 but you don't show any superscripts. and same for AN (g kg -1 TN) F-silage.
Same issue with Table 4 CP CP (g kg -1 DM) with p value of 0.032 for P-silage but no superscripts
Also check ADL (g kg -1DM) for both F-silage and P-silage with p-values of 0.11 and 0.009
Table 6 check IVDMD (g kg-1DM) F-silage with p-value of 0.034 and IVADFD (g kg-1DM) superscripts for F-silage.
Author Response
Dear reviewer, thank you for your encouraging and warm comments and suggestions, all of your suggestions are very important, and they all have important guiding significance for our future research work. Based on this we have revised and (we think) strengthened our paper.
Point 1: Line 17-18: Should it be High-water content of feed?
Response 1: Yes. Ensiling extends the storage life of feeds (with high water content).
Point 2: Line 25: why attractive? I suggest removing it
Response 2: We fully appreciate your suggestion. In the paper we have removed it.(PDF:25)
Point 3: Line 112: the experiment was conducted at the..........
Response 3: We fully appreciate your suggestion. We have reworked this section in the thesis. The experiment was conducted at the Animal Nutrition and Feed Science Experimental Base of Jilin University, Changchun City, Jilin Province, China (43.88°N, 125.35°E, altitude 300m).(PDF:116-118)
Point 4: Line 119 Table 1: I would have expected that you use the same raw materials in each spent mushroom substrate because differences could arise from using different raw materials which could affect the results for effectiveness of the mushroom type.
Response 4: We fully appreciate your suggestion. Using different raw materials in each waste mushroom substrate can make a difference, which can affect the results for effectiveness of the mushroom type. In subsequent trials, we will use the same raw material for each spent mushroom substrate as you suggest.
Point 5: Table 3: pH value for F-silage was significant p=0.002 but you don't show any superscripts. and same for AN (g kg -1 TN) F-silage. Same issue with Table 4 CP CP (g kg -1 DM) with p value of 0.032 for P-silage but no superscripts. Also check ADL (g kg -1DM) for both F-silage and P-silage with p-values of 0.11 and 0.009. Table 6 check IVDMD (g kg-1DM) F-silage with p-value of 0.034 and IVADFD (g kg-1DM) superscripts for F-silage.
Response5: We fully appreciate your suggestion. In Tables 3,4,6 we have rechecked the superscripts of the data.(PDF:280, 302, and 338)
Thank you again for your suggestions and hope to learn more from you.

Reviewer 2 Report (New Reviewer)
Animals-2074047-v1
General comments.
The study has a number of positive attributes. The authors attempt to evaluate the impacts of multiple additives on ensiling quality of 2 different mushroom residues. Their results vary greatly between the 2 different residues, thereby limiting the extrapolation of their treatments to other situations. This is a real strength of the work. Also, comparing both LAB and cellulase additions both alone and in combination provide interesting information, particularly when taken in the context of how the response differed between the different residues.
Another strength of the manuscript is the discussion section. The authors are to be commended for the depth of explanation they provide for their results and for how well they bring in studies to explain what should happen under different situations. The major problem with the discussion is that the authors often change the discussion to present tense for reporting previous results. All of the discussion should be past tense since that work has already been done.
On the negative side, the study is very complex in its design, implementation, and interpretation. The way the information is presented is very confusing and hard to follow. The authors themselves struggle with this, as in many instances, the authors incorrectly state a difference, or a lack of difference. This creates considerable confusion when trying to actually follow what the real impacts are. The confusion is compounded by reporting the analysis by main effects and interactions as well as using mean separations. The authors really do not do a good job specifying which they are discussing. They also confused me when they would lump multiple variables into one sentence. For example, they might state that multiple treatment effects were significant for IVDMD, IVNDFD, and IVHCD when that statement was not true for all of the variables. Please consider trying to simplify this in future versions of this work. The work is interesting, but the presentation is so confusing that it is difficult to determine what actually happened and what did not.
Individual comments are listed below.
Line Comment
16 suggest deleting “excellent”. We really don’t know unless research is done, and the digestibilities you report are not excellent. Also, if it is already done, then does this research really need to be conducted or does it really contribute to the field of knowledge?
34-36 This statement is incorrect. Ammonia of P silage was lower for L vs. Control but not different from E and M. Lactic acid was different in F silage between L, E, M and control and pH only differed in P silage between treatments and Control.
39-40 True for NDF digestibility but not for ADF.
42-45 These general statements cannot be made. They are not correct across all of the digestibility measurements. The varied responses between F and P silages in respect to the different additives makes it very difficult to generalize like this.
145 Most are using more automated procedures such as the ANKOM or other processes. Please specify that this was in fact conducted by refluxing individual samples in individual beakers.
175 suggest … fed to castrated rams
205 suggest … exposed to aerobic conditions…
224 Were NDF and ADF run sequentially on the same original sample? This should be specified.
238 You should mention the analysis of the aerobic stability data. You state that this was run on a mixed sample which implies that replicate silos were mixed together. Therefore, there is no replication. This should be clarified.
240-246 Most journals discourage repeating numbers that are presented in tables. The discussion should be centered around the relative differences, statistical differences, etc. For example, The WSC concentration from P-SMS was 1.4 times that of F-SMS….
257 Sentences cannot begin with an abbreviation.
257-259 This statement is not true across all treatments for either variable.
277-278 This statement is not true according to the p-values in Table 4.
279 There was an E treatment main effect but not an interaction for CP so this is incorrect.
280 True for E but not for L.
282-284 This seems like a statement that should begin the section rather than end it.
299 So were IVADFD.
300 Not true for IVADFD.
305-307 Again, this seems like it should be where the data discussion begins.
338 To me, it looks more like 10° rather than 2.
342 Why not mention L and E being lower than control?
363 The complete figure caption is not shown.
386 Yes, but your treatments did help the otherwise poor fermentation quality of F.
388 Not for all of the digestibility measurements.
409, 421, 467, etc. The sentences starts with an abbreviation.
413 Should there be citations here? You say, “in this experiment”.
468-469 I know we see it stated as the authors do here, but we are not digesting the animals. It should be digestion by beef cattle and digestibility by animals…
Discussion section. This section does a very thorough job of comparing the results to previous results. The major problem with this section is the inconsistency in use of present and past tense. For previous studies, the past tense should be used.
Tables. Most assign superscripts from greatest to least rather than least to greatest.
Author Response
Dear reviewer, thank you for your encouraging and warm comments and suggestions, all of your suggestions are very important, and they all have important guiding significance for our future research work. Based on this we have revised and (we think) strengthened our paper.
Point 1: 16 suggest deleting “excellent”. We really don’t know unless research is done, and the digestibilities you report are not excellent. Also, if it is already done, then does this research really need to be conducted or does it really contribute to the field of knowledge?
Response 1: We fully appreciate your suggestion. In the paper we have removed it. In China, the annual output of edible fungi is huge, and the two edible fungi with the largest output are Flammulina velutipes and Pleurotus eryngii. The yield of other edible fungi is low, and there is no prospect if they are used as feed, while the yield of Flammulina velutipes and Pleurotus eryngii is high and the source is stable. Therefore, how to deal with the residual SMS from the two edible fungi has become the focus of our research. We believe that this research can provide a theoretical basis for the production of high quality SMS fermented feeds.(PDF:16)
Point 2: 34-36 This statement is incorrect. Ammonia of P silage was lower for L vs. Control but not different from E and M. Lactic acid was different in F silage between L, E, M and control and pH only differed in P silage between treatments and Control.
Response 2: We fully appreciate your suggestion. In the thesis we have modified this statement. (PDF:34-36)
Point 3: 39-40 True for NDF digestibility but not for ADF.
Response 3: According to Table 6, compared with the control, the E treatment increased in vitro neutral detergent fibre digestibility (IVNDFD) and in vitro acid detergent fibre digestibility (IVADFD) in F-silage and P-silage (P<0.05).(PDF:338)
Point 4: 42-45 These general statements cannot be made. They are not correct across all of the digestibility measurements. The varied responses between F and P silages in respect to the different additives makes it very difficult to generalize like this.
Response 4: We fully appreciate your suggestion. Modified in the abstract section.(PDF:44-46)
Point 5: 145 Most are using more automated procedures such as the ANKOM or other processes. Please specify that this was in fact conducted by refluxing individual samples in individual beakers.
Response 5: We fully appreciate your suggestion. This was in fact conducted by refluxing individual samples in individual beakers.(PDF:154-155)
Point 6: 175 suggest … fed to castrated rams
Response 6: We fully appreciate your suggestion. fed to castrated rams. It has been revised in the dissertation.(PDF:185)
Point 7: 205 suggest … exposed to aerobic conditions…
Response 7: We fully appreciate your suggestion. exposed to aerobic conditions. It has been revised in the dissertation.(PDF:215)
Point 8: 224 Were NDF and ADF run sequentially on the same original sample? This should be specified.
Response 8: Yes. The NDF and ADF run sequentially on the same original sample.(PDF:234-235)
Point 9: 238 You should mention the analysis of the aerobic stability data. You state that this was run on a mixed sample which implies that replicate silos were mixed together. Therefore, there is no replication. This should be clarified.
Response 9: We fully appreciate your suggestion. In section 2.6 of the thesis, we have added an analysis of aerobic stability.(PDF:245)
Point 10: 240-246 Most journals discourage repeating numbers that are presented in tables. The discussion should be centered around the relative differences, statistical differences, etc. For example, The WSC concentration from P-SMS was 1.4 times that of F-SMS….
Response 10: We fully appreciate your suggestion. In section 3 of the paper, we have reworked. (PDF:252-254)
Point 11: 257 Sentences cannot begin with an abbreviation.
Response 11: We fully appreciate your suggestion. It has been revised in the dissertation.(PDF:275-278)
Point 12: 257-259 This statement is not true across all treatments for either variable.
Response 12: We fully appreciate your suggestion. It has been revised in the dissertation.(PDF:275-278)
Point 13: 277-278 This statement is not true according to the p-values in Table 4.
279: There was an E treatment main effect but not an interaction for CP so this is incorrect.
280:True for E but not for L.
Response 13: We fully appreciate your suggestion. It has been revised in the dissertation.(PDF:296-300)
Point 14: 282-284 This seems like a statement that should begin the section rather than end it.
Response 14: We fully appreciate your suggestion. It has been revised in the dissertation.(PDF:292-295)
Point 15: 299 So were IVADFD.
300 Not true for IVADFD.
Response 15: We fully appreciate your suggestion. It has been revised in the dissertation.(PDF:323)
Point 16: 305-307 Again, this seems like it should be where the data discussion begins.
Response 16: We fully appreciate your suggestion. It has been revised in the dissertation.(PDF:316-321)
Point 17: 338 To me, it looks more like 10° rather than 2.
Response 17: We have rechecked the data on the aerobic stability of the P-silage. At 132 h, the temperature in the control group was 2°C higher than the ambient temperature. At 150 h, the temperature in the M treatment group was 2°C higher than the ambient temperature.
Point 18: 342 Why not mention L and E being lower than control?
Response 18: We fully appreciate your suggestion. It has been revised in the dissertation.(PDF:360-363)
Point 19: 363 The complete figure caption is not shown.
Response 19: We fully appreciate your suggestion. It has been revised in the dissertation.(PDF:384)
Point 20: 386 Yes, but your treatments did help the otherwise poor fermentation quality of F.
Response 20: In general, high-quality silage has a pH of 3.8-4.2, at which time the activity of harmful bacteria in the silage is attenuated. F-silage in the 4 treatment groups exhibited a poor fermentation quality, as reflected in the high pH (4.2-4.4) and BA (5.0 g kg-1 DM -11.2 g kg-1 DM). Although the pH in F-silage from the L, E, and M groups were lower than those in the control group.
Point 21: 388 Not for all of the digestibility measurements.
Response 21: We fully appreciate your suggestion. It has been revised in the dissertation.(PDF:408-410)
Point 22: 409, 421, 467, etc. The sentences starts with an abbreviation.
Response 22: We fully appreciate your suggestion. It has been revised in the dissertation.(PDF:401)
Point 23: 413 Should there be citations here? You say, “in this experiment”.
Response 23: We fully appreciate your suggestion. It has been revised in the dissertation.(PDF:433-436)
Point 24: 468-469 I know we see it stated as the authors do here, but we are not digesting the animals. It should be digestion by beef cattle and digestibility by animals…
Response 24: We fully appreciate your suggestion. In the paper we have removed it.(PDF:496-499)
Point 25: Discussion section. This section does a very thorough job of comparing the results to previous results. The major problem with this section is the inconsistency in use of present and past tense. For previous studies, the past tense should be used.
Response 25: We fully appreciate your suggestion. And we have revised the discussion section. (PDF:401)
Thank you again for your suggestions and hope to learn more from you.

Reviewer 3 Report (New Reviewer)
Title
ok
Abstract
Ok. It is suggested to used better abbreviations that allow an easy reading
Introduction
ok
Materials and Methods
Line 112. Please revise redaction: “ . . . experimental site of this experiment is . . .”
Line 114. Please complete: …altitude 300 m (above sea level).
Line 116: Please revise “ … removing edible parts from the edible … “
Line 125: Please revise: “… was measured as 4.2 × 10-3 U g-1 FW …”
Line 130: “… it was put into a 5 L polyethene plastic bucket with an …” Please indicate how many 5-L- polyethene-plastic-bags were used per treatment.
Line 135: You indicated that “ … Five replicates per treatment were used for the in vitro digestibility analysis…” Do you mean you used five 5-L- polyethene-plastic-bags per treatment?
It is not clear how the statistical analysis of variance was undertaken. Could you please include de statistical model(s) used. This could help to make it clear.
Results
Lines 240 – 247. Most part of this section is repeating data already indicated in the Table 2. Please, find a better way to describe the results; eg. Indicating the items that were higher for one type of silage than the other; also indicating the difference in percentage.
In Table 3, you show that pH in the P-silage was lower in the L, E and M that the control, whereas pH of F-silage was not affected by L, E and M. However, the content of LA was in the other way around, and LA could be one of the main factors that reduce pH. Please revise the data.
Line 261. Please revise: L and E treatments influenced pH and AN/TN in F-silage (P=0.002-0.048).
Line 277. “ … three additional treatments… “ What do you mean?
Discussion
Lines 410 - 412. Please revise: “In this experiment, compared with the control group, the addition of L. plantarum accelerated the accumulation of LA, decreased the pH and AN/TN ratio, and improved the fermentation quality of P-silage” Table 3 shows that LA was not significantly (P-value= 0.839 ) affected by treatments in P-silage.
Lines 419 – 420. Please revise “… pH reduction is stronger than that of L. plantarum alone, which is consistent with the results of this experiment … “. Table 3 shows no difference between L, E and M treatments in P-silage.
Lines 521 – 523. Please revise “Although the IVDMD, IVNDFD, and IVADFD of F-silage in the E and M treatment groups were higher than those in the control treatment group, indicating that the digestibility of nutrients in sheep was improved, the fermentation quality of F-silage was very poor.” It is unclear.
Tables
Table 1. Please correct: Wheat brain ; Wood flour : sawdust
Table 3. Item pH value, F-silage has a P-value of 0.002 but, Tukey test was not undertaken.
Table 3. Item AN, F-silage and P-silage, with P-value of 0.023 and 0.007, Tukey test was not undertaken.
Table 4. CP in P-silage has a P-value of 0.032, but not Tukey tests was made. This is also the case for ADL, for both F-silage and P-silage (0.011 and 0.009)
Table 5. IVDMD in F-silage has a P-value of 0.34, Tukey test was not undertaken
Figures
Figure 1. Revise the title
Author Response
Dear reviewer, thank you for your encouraging and warm comments and suggestions, all of your suggestions are very important, and they all have important guiding significance for our future research work. Based on this we have revised and (we think) strengthened our paper.
Point 1: Line 112. Please revise redaction: “ . . . experimental site of this experiment is . . .”
Response 1: We fully appreciate your suggestion. It has been revised in the dissertation.(PDF:116-118)
Point 2: Line 114. Please complete: …altitude 300 m (above sea level).
Response 2: We fully appreciate your suggestion. It has been revised in the dissertation.(PDF:118)
Point 3: Line 116: Please revise “ … removing edible parts from the edible … “
Response 3: We fully appreciate your suggestion. It has been revised in the dissertation.(PDF:120-123)
Point 4: Line 125: Please revise: “… was measured as 4.2 × 10-3 U g-1 FW …”
Response 4: We fully appreciate your suggestion. It has been revised in the dissertation.(PDF:129-138)
Point 5: Line 130: “… it was put into a 5 L polyethene plastic bucket with an …” Please indicate how many 5-L- polyethene-plastic-bags were used per treatment.
Response 5: Each treatment was designed with 5 replicates and five 5-L polythene-buckets were used. (PDF:140-141)
Point 6: Line 135: You indicated that “ … Five replicates per treatment were used for the in vitro digestibility analysis…” Do you mean you used five 5-L- polyethene-plastic-bags per treatment?
Response 6: Yes,Each treatment was designed with 5 replicates and five 5-L polythene-buckets were used.
Point 7: Lines 240 – 247. Most part of this section is repeating data already indicated in the Table 2. Please, find a better way to describe the results; eg. Indicating the items that were higher for one type of silage than the other; also indicating the difference in percentage.
Response 7: We fully appreciate your suggestion. In section 3 of the paper, we have reworked. (PDF:252-254)
Point 8: In Table 3, you show that pH in the P-silage was lower in the L, E and M that the control, whereas pH of F-silage was not affected by L, E and M. However, the content of LA was in the other way around, and LA could be one of the main factors that reduce pH. Please revise the data.
Response 8: We fully appreciate your suggestion. In Table 3, we have reworked it. The pH in F-silage and P-silage from the L, E, and M groups were lower than those in the control group.
Point 9: Line 261. Please revise: L and E treatments influenced pH and AN/TN in F-silage (P=0.002-0.048).
Response 9: We fully appreciate your suggestion. It has been revised in the dissertation.(PDF:271-272)
Point 10: Line 277. “ … three additional treatments… “ What do you mean?
Response 10: We fully appreciate your suggestion. The DM content in P-silage from the L, E, and M groups were higher than that in the control group (P<0.05). We have modified the expression of the sentence.(PDF:296-297)
Point 11: Lines 410 - 412. Please revise: “In this experiment, compared with the control group, the addition of L. plantarum accelerated the accumulation of LA, decreased the pH and AN/TN ratio, and improved the fermentation quality of P-silage” Table 3 shows that LA was not significantly (P-value= 0.839 ) affected by treatments in P-silage.
Response 11: We fully appreciate your suggestion. It has been revised in the dissertation.(PDF:433-436)
Point 12: Lines 419 – 420. Please revise “… pH reduction is stronger than that of L. plantarum alone, which is consistent with the results of this experiment … “. Table 3 shows no difference between L, E and M treatments in P-silage.
Response 12: We fully appreciate your suggestion. It has been revised in the dissertation.(PDF:444-447)
Point 13: Lines 521 – 523. Please revise “Although the IVDMD, IVNDFD, and IVADFD of F-silage in the E and M treatment groups were higher than those in the control treatment group, indicating that the digestibility of nutrients in sheep was improved, the fermentation quality of F-silage was very poor.” It is unclear.
Response 13: We fully appreciate your suggestion. We did not express this sentence clearly enough and have revised it in the paper.(PDF:549-558)
Point 14: Table 1. Please correct: Wheat brain ; Wood flour : sawdust
Response 14: We fully appreciate your suggestion. We have corrected this in Table 1.(PDF:128)
Point 15: Table 3. Item pH value, F-silage has a P-value of 0.002 but, Tukey test was not undertaken. Table 3. Item AN, F-silage and P-silage, with P-value of 0.023 and 0.007, Tukey test was not undertaken. Table 4. CP in P-silage has a P-value of 0.032, but not Tukey tests was made. This is also the case for ADL, for both F-silage and P-silage (0.011 and 0.009). Table 6. IVDMD in F-silage has a P-value of 0.34, Tukey test was not undertaken
Response 15: We fully appreciate your suggestion. In Tables 3,4,6 we have rechecked the superscripts of the data.(PDF:280, 302, and 338)
Point 16: Figure 1. Revise the title
Response 16: We fully appreciate your suggestion. It has been revised in the dissertation.(PDF:384)
Thank you again for your suggestions and hope to learn more from you.

Round 2
Reviewer 2 Report (New Reviewer)
Animals-2074047 Revision 1
The authors made considerable progress in getting the manuscript closer to publishable quality in a short timeframe. I commend them for their efforts. The authors have successfully addressed most of the issues that I found with the manuscript. However, there are still issues that I list below that I feel they should address. One of these issues is how the treatments are compared. The authors switch back and forth between comparisons of main effect and comparison of pair-wise comparisons. This gets really confusing. I would suggest that they begin talking about the overall effects of L, E and M and then get into the pairwise comparisons when the interaction was significant. I would also suggest that when they are talking about the main effects of L or E, they should state that specifically to avoid confusion.
As I elude to toward the end of my comments, I feel the authors have the mindset that if the silage does not meet criteria for excellent silage, then it is a waste of time. This simply is not the case in my opinion. A number of scientists have tried to develop prediction equations to estimate which silage chemical components to use to best predict intake or animal performance. Their overall prediction equations were not strong, indicating that consumption of silages is highly variable and preference is based on a myriad of contributing factors. Therefore, I caution the authors in being too hard on the F silage since it didn’t meet ideal conditions and focus on whether improvements were made with the different additives.
Overall, the research has merit and represents good work. I am just trying to make the presentation of the information better so the simple-minded like myself can understand it more clearly.
Additional comments are as follows.
34-36 This is still not correct for AN/TN. The L and E treatments within F and the E and M treatments within P are not different from the control based on the “b” superscript overlap.
142-143 The way this is written is awkward. You answered my question about the individual beakers. I think the wording in parentheses can be deleted. I apologize for the confusion.
142 I would suggest adding the following … were determined sequentially using the … This will eliminate the question about hemicellulose digestibility on line 222 in the revision.
172 The correction … fed to castrated rams… was not made as intended.
222 This is not necessary if the above comment is adhered to (line 142).
240 suggest… The DM content of P-SMS was 12 g kg-1 greater than that of F-SMS…
259-260 This statement still isn’t correct for AN/TN. The superscripts overlap.
275-276 This is confusing since the superscripts overlap. Maybe if you would state that the main effect of E influenced…
280-282 This is true only for ADF based on the superscripts in Table 4.
296 As mentioned before, the IVADFD were also greater from F than from P.
330-336 This discussion is about P silages or Figure 2. If this is only 2 degrees of change, the axis is wrong. Treatments L, M, and E all peak above 32 degrees according to the figure and ambient temperature stayed around 24.
387-388 I would still argue that it should be noted that each of your treatments did improve fermentation of F silages. I understand your point about standards for good silage. However, some feedstuffs simply do not ferment well and any help is better than no help. Also, as a side comment, the more I work with silage, the less concerned I get about silage quality standards and more about simply preserving the feedstuff. In one study, animals actually consumed very poorly fermented silage over silage that would be considered better quality silage, likely because of greater sugar content of the poorly-fermented silage. That was a different situation than here, but I still think that considering the poor relationship between silage quality measurements and consumption by animals, it is noteworthy to point out the improvements you made in F silages.
524 This sentence should be changed to past tense.
Author Response
Dear reviewer, thank you for your encouraging and warm comments and suggestions, all of your suggestions are very important, and they all have important guiding significance for our future research work. Based on this we have revised and (we think) strengthened our paper.
Point 1: 34-36 This is still not correct for AN/TN. The L and E treatments within F and the E and M treatments within P are not different from the control based on the “b” superscript overlap.
Response 1: We fully appreciate your suggestion. It has been revised in the dissertation.(PDF:35)
Point 2: 142-143 The way this is written is awkward. You answered my question about the individual beakers. I think the wording in parentheses can be deleted. I apologize for the confusion.
Response 2: We fully appreciate your suggestion. In the paper we have removed it.(PDF:142-145)
Point 3: 142 I would suggest adding the following … were determined sequentially using the … This will eliminate the question about hemicellulose digestibility on line 222 in the revision.
Response 3: We fully appreciate your suggestion. The NDF and ADF run sequentially on the same original sample. The NDF and ADF were determined sequentially using neutral detergent and acid detergent respectively.(PDF:142-145)
Point 4: 172 The correction … fed to castrated rams… was not made as intended.
Response 4: We fully appreciate your suggestion. It has been revised in the dissertation.(PDF:173-174)
Point 5: 240 suggest… The DM content of P-SMS was 12 g kg-1 greater than that of F-SMS…
Response 5: We fully appreciate your suggestion. It has been revised in the dissertation.(PDF:241-242)
Point 6: 259-260 This statement still isn’t correct for AN/TN. The superscripts overlap.
Response 6: We fully appreciate your suggestion. It has been revised in the dissertation.(PDF:262-263)
Point 7: 275-276 This is confusing since the superscripts overlap. Maybe if you would state that the main effect of E influenced…
Response 7: We fully appreciate your suggestion. It has been revised in the dissertation.(PDF:281-283)
Point 8: 280-282 This is true only for ADF based on the superscripts in Table 4.
Response 8: We fully appreciate your suggestion. It has been revised in the dissertation.(PDF:286-289)
Point 9: 296 As mentioned before, the IVADFD were also greater from F than from P.
Response 9: We fully appreciate your suggestion. It has been revised in the dissertation.(PDF:306)
Point 10: 330-336 This discussion is about P silages or Figure 2. If this is only 2 degrees of change, the axis is wrong. Treatments L, M, and E all peak above 32 degrees according to the figure and ambient temperature stayed around 24.
Response 10: We fully appreciate your suggestion. We rechecked the aerobic stability data and found that the description in that paragraph was inaccurate and then revised the paragraph. Figures 1 and 2 show the change in all temperatures for both silages over 312 h, not just the 2°C change. The basis for determining aerobic stability: Aerobic stability was determined based on the time required for the silage temperature to be 2°C higher than the ambient temperature.(PDF:338-343)
Point 11: 387-388 I would still argue that it should be noted that each of your treatments did improve fermentation of F silages. I understand your point about standards for good silage. However, some feedstuffs simply do not ferment well and any help is better than no help. Also, as a side comment, the more I work with silage, the less concerned I get about silage quality standards and more about simply preserving the feedstuff. In one study, animals actually consumed very poorly fermented silage over silage that would be considered better quality silage, likely because of greater sugar content of the poorly-fermented silage. That was a different situation than here, but I still think that considering the poor relationship between silage quality measurements and consumption by animals, it is noteworthy to point out the improvements you made in F silages.
Response 11: We fully appreciate your suggestion. In this trial we focused too much on the quality standard of the silage and ignored the fact that each treatment did improve the fermentation quality of the F-silage. We strongly agree with your view that some feedstuffs simply do not ferment well and any help is better than no help. The more I work with silage, the less concerned I get about silage quality standards and more about simply preserving the feedstuff. It has been revised in the dissertation.(PDF:399-402)
Point 12: 524 This sentence should be changed to past tense.
Response 12: We fully appreciate your suggestion. It has been revised in the dissertation.(PDF:540)
Thank you again for your suggestions and hope to learn more from you.

This manuscript is a resubmission of an earlier submission. The following is a list of the peer review reports and author responses from that submission.
Round 1
Reviewer 1 Report
This work present novel informations and huge amounts of data for ensiling spent mushroom substrate. The authors investigated the fermentation quality, in vitro digestibility, and aerobic stability of Flammulina velutipes spent mushroom substrate silage and Pleurotus eryngii spent mushroom substrate silage. The results was interesting, the words must be improved.
This article mainly studied the silage technology of mushroom matrix and the influence of additives on it This article has a large amount of data and has achieved novel results. I think it can attract readers This article is innovative because it compares the characteristics of two mushroom substrates after silage and finds that different additives are suitable for silage modulated by different substrates, which is not found so far However, the English writing of this article needs to be improved. The author can turn to English polishing institutions or native English speakers to modify the language The data in this paper can support the conclusion obtained. The author also made a detailed study and analysis on the quality and influencing factors of silage made from different mushroom substrates
Author Response
Dear reviewer, thank you for your encouraging and warm comments and suggestions, all of your suggestions are very important, and they all have important guiding significance for our future research work. Based on this we have revised and (we think) strengthened our paper.
Point 1: The English writing of this article needs to be improved. The author can turn to English polishing institutions or native English speakers to modify the language
Response 1: We fully appreciate your suggestion. We have turned to an English polishing agency and done English polishing.
Thank you again for your suggestions and hope to learn more from you.

Author Response
Dear reviewer, thank you for your encouraging and warm comments and suggestions, all of your suggestions are very important, and they all have important guiding significance for our future research work. Based on this we have revised and (we think) strengthened our paper.
Point 1: this phrase is not correct. reformulate it please
Response 1: We fully appreciate your suggestion. We have rephrased the sentence.(PDF:13-15)
Point 2: avoid to liest the results in simple summary but report implications for the practical point of view
Response 2: We fully appreciate your suggestion. Results are reported from a practical point of view, modified in the Simple Summary section. (PDF:23-26)
Point 3: add a short phrase about statistics in the abstract
Response 3: We fully appreciate your suggestion. A phrase about statistics has been added to the abstract. For example, data analysis was performed using independent sample T test and analysis of variance. (PDF:34-35)
Point 4: add more structured conclusions in the abstract
Response 4: We fully appreciate your suggestion. The conclusions in the abstract should be more structured. (PDF:46-50)
Point 5: The introduction is not clearly written, pleas refer to a lenguage review service
Response 5: We fully appreciate your suggestion. In the paper, we have added an introduction section. (PDF:55-62)
Point 6: I suggest to start the paper with importane of by products in livestock nutrition, as example I suggest to include different.
Response 6: We fully appreciate your suggestion. In the introductory section, we have added information on the importance of crop by-products for livestock nutrition. (PDF:55-62)
Point 7: The hypotesis of the study is missing
Response 7: We fully appreciate your suggestion. We added research hypotheses in the introduction section. (PDF:100-104)
Point 8: usually the m&m section start with the location of the sutdy and hetical permission (if required)
Response 8: We fully appreciate your suggestion. In the m&m section, we have added the study site and ethical approval. (PDF:110-112,205-207)
Point 9: How did you chose the number of replicates? report in the text
Response 9: We choose the number of repetitions this way. Repetition is one of the important principles of experimental design, and experimental repetition plays a decisive role in the reproducibility of experimental results and the reliability of final experimental conclusions. In the experimental design, repeated experiments can be set to estimate the experimental error; the more repeated experiments, the lower the experimental error and the higher the experimental precision. Therefore, in order to have an adequate sample size for silage evaluation and to provide more robustness to the experimental data, we performed five replicates in each treatment group. At the same time, we also refer to the experimental design of Chen et al, and each treatment group takes five repetitions (https://doi.org/10.1111/grs.12240).
Point 10: before to select the test did you perform distribution and homoscedasticity analysis, as reported in
Response 10: We fully appreciate your suggestion. Before choosing a test, we performed a test for distribution and homoscedasticity analysis. The normal distribution test and the homogeneity test were performed before performing ANOVA, and the normal distribution test was performed before performing the independent samples T-test.
Point 11: when you have only 2 treatments letters are not necessary
Response 11: We fully appreciate your suggestion. We have removed unnecessary letters from the table.
Point 12: This p value what is regardind to? please report a foot note
Response 12: The P value in the same row represents the P value of the analysis of variance, and different lowercase letters (a, b) in the same row indicate that there is a significant difference between different additive treatment groups under the same SMS (P<0.05); The P value in the same column represents the P value of the independent sample T test. If p < 0.05, it means that there is a significant difference between different SMS in the same additive treatment group. Footnotes have been added to the table.
Point 13: I suggest to report diffrent letters, such as x, y and z
Response 13: We fully appreciate your suggestion. We've removed capital letters from the same column in the table. Different lowercase letters (a, b and c) in the same line indicate that there is a significant difference between different additive treatment groups under the same spent mushroom substrate (P<0.05).
Point 14: Report in the figure the full model p values
Response 14: We fully appreciate your suggestion. Figure 3 and Figure 4 show the effects of different additives on the aerobic stability of F-silage and P-silage exposed to air, respectively. The homogeneity of variance test was carried out before the one-way ANOVA, and the conclusion of homogeneity of variance was obtained, which met the applicable conditions of the ANOVA. A one-way ANOVA was then performed with Tukey's method for multiple comparisons. Different lowercase letters in the figure indicate significant differences (P<0.05) among different additive treatment groups. We drew Figures 3 and 4 with reference to the presentation of Bernardi et al's paper on the effect of different additives on the aerobic stability of silage exposed to air (https://doi.org/10.1111/gfs.12452). At the same time, we have added the F and P values of the one-way ANOVA to footnotes. (PDF:350,355)
Point 15: sually the discussion start with repeating the hypotesis
Response 15: We fully appreciate your suggestion, and the discussion starts with repeating the hypothesis. Regarding the fermentation quality of silage, the hypothesis of the study was that the combined application of lactic acid bacteria and cellulase could increase the lactic acid content of SMS silage, thereby improving the fermentation quality of SMS silage. In fact, P-silage obtained good fermentation. F-silage in the 4 treatment groups exhibited a poor fermentation quality, as reflected in the high pH (4.2-4.4) and BA (5.0 g kg-1 DM -11.2 g kg-1 DM), despite the higher LA content than well-preserved silages (30 g kg -1 DM). It can be seen that the speculation about the fermentation quality of F-silage is not in line with the reality. (PDF:361-366)
Point 16: I suggest to add a section explaining the possible utilization of this silage in animal nutrition
Response 16: We fully appreciate your suggestion. We have added a section to the Discussion section explaining possible utilization of P-silage in animal nutrition. The L-treated P-silage is introduced into ruminants diet, it may maximize the use of agricultural byproduct, promote the balance of ruminants diet, increase food intake in ruminants, and improve the production performance of ruminants. (PDF:494-500)
Thank you again for your suggestions and hope to learn more from you.

Reviewer 3 Report
Dear authors, ensiling spent mushroom substrate as a feed source for ruminant seems to be interesting, the manuscript was well prepared in both experimental design and data analysis, however, the manuscript still need some improvement.
Line 184, the detail of the filter bag should be added
Line 192-194, how did you determine IVDMD, IVCPD, IVOMD and IVGED? And why you did not determine IVNDFD and INADFD?
Line 196, the information of SPSS should be revised (version, company, city, province, country)
The upper case letters in the same column in tables should be delete
I think the main purpose of this study was to select the most optimal treatment for P-silage and F-silage, however, the difference between P-silage and F-silage should be less important, thus, the results of P- and F-silage were suggested to be showed separately, to make the tables more readable.
Author Response
Dear reviewer, thank you for your encouraging and warm comments and suggestions, all of your suggestions are very important, and they all have important guiding significance for our future research work. Based on this we have revised and (we think) strengthened our paper.
Point 1: Line 184, the detail of the filter bag should be added
Response 1: We fully appreciate your suggestion. In section 2.5, we added details of the filter bag (ANKOM F57; aperture 25um; ANKOM Technology Corporation; Macedon, NY, USA). (PDF:208-209)
Point 2: Line 192-194, how did you determine IVDMD, IVCPD, IVOMD and IVGED? And why you did not determine IVNDFD and INADFD?
Response 2:
We fully appreciate your suggestion. In the design of the in vitro digestibility test, we refer to the test design of Gao et al to determine IVDMD, IVCPD, IVOMD, and IVGED, but not IVNDFD and INADFD (https://doi.org/10.5713/ajas.18.0886). At the same time, on the one hand, we consider it from the perspective of beef cattle fattening, from the perspective of protein, organic matter and total energy, and do not consider NDF and ADF aspects; On the other hand, fiber energy is part of the total energy, and we want to embody the NDF and ADF by the total energy. Therefore, only IVDMD, IVCPD, IVOMD, and IVGED were determined, but not IVNDFD and INADFD.
The purpose of this experiment is to carry out the industrialized production of SMS fermented feed, and it is a continuous experiment. In a subsequent in vitro digestibility test, we will follow your advice and determine IVNDFD and INADFD. We fully agree with your point of view, and once again express our thanks to you.
Point 3: Line 196, the information of SPSS should be revised (version, company, city, province, country)
Response 3: We fully appreciate your suggestion. We have revised the SPSS information (version 26; International Business Machines Corporation; Armonk, New York, USA). (PDF:220-221)
Point 4: The upper case letters in the same column in tables should be delete.
Response 4: We fully appreciate your suggestion. We've removed capital letters from the same column in the table.
Point 5: I think the main purpose of this study was to select the most optimal treatment for P-silage and F-silage, however, the difference between P-silage and F-silage should be less important, thus, the results of P- and F-silage were suggested to be showed separately, to make the tables more readable.
Response 5: We fully appreciate your suggestion. The main purpose of this study was to select the optimal treatment for F-silage and P-silage. From the conclusion, we can see that the combination of application of lactic acid bacteria and cellulase is an effective strategy for improving the fermentation quality and aerobic stability of P-silage. Adding lactic acid bacteria or cellulase alone improves the aerobic stability of F-silage. Lactic acid bacteria can be used as an attractive additive to improve the in vitro digestibility of P-silage.
In China, the annual output of edible fungi is huge, and the two edible fungi with the largest output are Flammulina velutipes and Pleurotus eryngii. The yield of other edible fungi is low, and there is no prospect if they are used as feed, while the yield of Flammulina velutipes and Pleurotus eryngii is high and the source is stable. Therefore, how to deal with the residual SMS from the two edible fungi has become the focus of our research. To provide a theoretical basis for the production of high-quality SMS fermented feed, the effects of lactic acid bacteria and cellulase on the fermentation quality, in vitro digestibility, and aerobic stability in the industrially produced SMS silage were investigated in this experiment.
The table design of this experiment also refers to the table of Cai et al's paper, using lactic acid bacteria inoculant and cellulase to prepare corn straw silage and sugarcane straw silage. The results of the two silages are displayed in the same table, to find out which raw material is more suitable for ensiling and which additives can be added to obtain high-quality silage (https://doi.org/10.5713/ajas.19.0348). The main purpose of putting the silage results of F-silage and P-silage in a table is to intuitively compare the silage effects of the two SMS, and to understand which SMS is more suitable for silage fermentation, so as to lay a good foundation for the later beef cattle fattening experiments, and get higher economic benefits in future industrial production. The experimental results are obvious, the Pleurotus eryngii SMS treated with lactic acid bacteria is more suitable for silage fermentation and has better fermentation quality. Therefore, in the later beef cattle fattening test, we will feed the P-silage inoculated with lactic acid bacteria to explore its effect on beef cattle production performance, slaughter performance and fattening benefits.
Thank you again for your suggestions and hope to learn more from you.

Round 2
Reviewer 2 Report
I really appreciate if the authors check again the point 6 and 15 and implemented the answers in the text adding the missing suggestions
Author Response
Dear reviewer, thank you for your encouraging and warm comments and suggestions, all of your suggestions are very important, and they all have important guiding significance for our future research work. Based on this we have revised and (we think) strengthened our paper.
Point 1: I really appreciate if the authors check again the point 6 and 15 and implemented the answers in the text adding the missing suggestions
Response 1:
(1) Point 6: I suggest to start the paper with importane of by products in livestock nutrition, as example I suggest to include different.
In the introductory section, we have added information on the importance of crop by-products for livestock nutrition. (PDF:63-70)
(2) Point 15: sually the discussion start with repeating the hypotesis
We fully appreciate your suggestion, and the discussion starts with repeating the hypothesis. (PDF:372-382)
Thank you again for your suggestions and hope to learn more from you.

Reviewer 3 Report
The manuscript was carefully revised in the current version, however, I still have some concerns about the sample analysis and the description of the results.
Firstly, as showed in Table 2, the CP content was quite low, whereas the NDF and ADF was great, thus, both of these two silages should be kind of fiber source for ruminant, and I believe the in vitro digestibility of NDF, ADF and hemicellulose should be more valuable than CP, OM and GE. Moreover, the filter bag you used in the current study was also fits the Ankom fiber analyzer (DELTAi), thus, the analysis of NDF and ADF could be easy and accurate. And, since the reference (19) you mentioned did not mention the analysis of CP, OM and GE, I do not know how did you analyze these indicators. But the result should not be acceptable if you cut the filter bag, and take the feed out for analysis, since the feed always stick to the bags, and thus the feed samples you got could not be representative. I strongly suggest the authors to remove the digestibility of CP, OM and GE, and add the digestibility of NDF, ADF and hemicellulose.
Secondly, as showed in Table 1, the composition of these two substrates was quite different, it’s not meaningful to compare the fermentation quality, chemical composition, and in vitro digestibility between these two silages. Furthermore, the objective of this study you mentioned in the introduction section was also to explore the optimal treatment for different silages, not to explore which one is suitable for silage fermentation. Thus, I still strongly suggest the authors to show the results of these two silages separately.
Author Response
Dear reviewer, thank you for your encouraging and warm comments and suggestions, all of your suggestions are very important, and they all have important guiding significance for our future research work. Based on this we have revised and (we think) strengthened our paper.
Point 1: Firstly, as showed in Table 2, the CP content was quite low, whereas the NDF and ADF was great, thus, both of these two silages should be kind of fiber source for ruminant, and I believe the in vitro digestibility of NDF, ADF and hemicellulose should be more valuable than CP, OM and GE. Moreover, the filter bag you used in the current study was also fits the Ankom fiber analyzer (DELTAi), thus, the analysis of NDF and ADF could be easy and accurate. And, since the reference (19) you mentioned did not mention the analysis of CP, OM and GE, I do not know how did you analyze these indicators. But the result should not be acceptable if you cut the filter bag, and take the feed out for analysis, since the feed always stick to the bags, and thus the feed samples you got could not be representative. I strongly suggest the authors to remove the digestibility of CP, OM and GE, and add the digestibility of NDF, ADF and hemicellulose.
Response 1: We fully appreciate your suggestion. In vitro digestibility of NDF, ADF and hemicellulose should be more valuable than that of CP, OM and GE. Therefore, we removed the digestibility of CP, OM and GE, and added the digestibility of NDF, ADF and hemicellulose (PDF:313). At the same time, we reviewed the references and revised the determination methods of the above indicators (PDF:219-228).
Point 2: Secondly, as showed in Table 1, the composition of these two substrates was quite different, it’s not meaningful to compare the fermentation quality, chemical composition, and in vitro digestibility between these two silages. Furthermore, the objective of this study you mentioned in the introduction section was also to explore the optimal treatment for different silages, not to explore which one is suitable for silage fermentation. Thus, I still strongly suggest the authors to show the results of these two silages separately.
Response 2: We fully appreciate your suggestion. In the result part of the paper, we show the results of F-silage and P-silage respectively to make the table more readable. (PDF: 255)
Thank you again for your suggestions and hope to learn more from you.
